

# Scotland's Forgotten Carbon: A National Assessment of Mid-Latitude Fjord Sedimentary Carbon Stocks.

Craig Smeaton [1,*], William E.N. Austin [1,2], Althea L. Davies [1], Agnes Baltzer [3], John A. Howe [2] and John M. Baxter [4].

[1] School of Geography & Geosciences, University of St Andrews, St Andrews, KY16 9AL, UK

[2] Scottish Association for Marine Science, Scottish Marine Institute, Oban, PA37 1QA, UK

[3] Institut de Géographie et d'Aménagement Régional de l'Université de Nantes, BP 81 227 44312 Nantes CEDEX 3, France.

[4] Scottish Natural Heritage, Silvan House, Edinburgh, EH12 7AT, UK

**\*Corresponding Author.**

Email address: cs244@st-andrews.ac.uk (Craig Smeaton)

**Highlights:**

- Scottish fjords are a more effective store of C than the terrestrial environment.
- A total of 640.7 ± 46 Mt C is stored in the sediment of Scotland's 111 fjords.
- An estimated 31139-40615 t yr$^{-1}$ C is buried in the sediment of Scotland's fjords.
- Fjord sediments are potentially the most effective store of C globally.



**Abstract**
Fjords are recognised as hotspots for the burial and long-term storage of carbon (C) and
potentially provide a significant climate regulation service over multiple timescales.
Understanding the magnitude of marine sedimentary C stores and the processes which govern
their development is fundamental to understanding the role of the coastal ocean in the global
C cycle.  In this study, we use the mid-latitude fjords of Scotland as a natural laboratory to
further develop methods to quantify these marine sedimentary C stores at both the individual
fjord and national scale. Targeted geophysical and geochemical analysis has allowed the
quantification of sedimentary C stocks for a number of mid-latitude fjords and, coupled with
upscaling techniques based on fjord classification, has generated the first full national
sedimentary C inventory for a fjordic system. The sediments within these mid-latitude fjords
hold 640.7 ± 46 Mt of C split between 295.6 ± 52 and 345.1 ± 39 Mt of organic and inorganic
C respectively. When compared, these marine mid-latitude sedimentary C stores are of
similar magnitude to their terrestrial equivalents, with the exception of the Scottish peatlands,
which hold significantly more C. However, when area-normalised comparisons are made,
these mid-latitude fjords are significantly more effective as C stores than their terrestrial
counterparts, including Scottish peatlands. The C held within Scotland's coastal marine
sediments has been largely overlooked as a significant component of the nation's natural
capital; such coastal C stores are likely to be key to understanding and constraining improved
global C budgets.


















## 1. Introduction

Globally there is growing recognition that the burial (Smith et al., 2015) and storage (Smeaton et al., 2016) of carbon (C) in coastal marine sediments is an important factor in the global carbon cycle (Bauer et al. 2013), as well as providing an essential climate regulating service (Smith et al. 2015). Coastal sediments have been shown to be globally significant repositories for C, with an estimated 126.2 Mt of C being buried annually (Duarte et al., 2005). Of the different coastal depositional environments, fjords have been shown to be 'hotspots' for C burial, with approximately 11 % of the annual global marine carbon sequestration occurring within fjordic environments (Smith et al., 2015). Although it is clear these areas are important for the burial and long-term storage of C, the actual quantity of C held within coastal sediment remains largely unaccounted for. This knowledge deficit hinders our ability to fully evaluate, manage and protect these coastal C stores and the climate-regulating service that they provide.

The quantification of C in fjordic sediments was identified as a priority by Syvitski et al. (1987), but little progress has been made towards this goal until recently. Our work presented here utilises and extends the joint geochemistry and geophysical methodology developed by Smeaton et al. (2016) by applying it to a number of mid-latitude fjords. Estimated sedimentary C stocks for individual fjords will be utilised to create the first national estimate of sedimentary C stocks in the coastal ocean and thus quantify an overlooked aspect of Scotland's natural capital.

## 2. Scotland's Fjords

The coastal landscape of the west coast and islands of Scotland is dominated by fjordic geomorphology (Cage and Austin, 2010; Nørgaard-Pedersen et al., 2006). Catchments totalling an area of 21,742 km$^2$ drain to the sea through fjords, thus transporting sediment from the C rich soils into the marine system (Bradley et al., 2005). There are 111 large fjords (over 2 km long, where fjord length is twice fjord width) (Fig.1) in Scotland (Edwards and Sharples, 1986), supplemented by a further 115 smaller systems. The 111 large fjords are the primary focus of this study because their size and heavily glaciated geomorphology (Howe et al., 2002) suggest they are likely to store significant quantities of postglacial sediment. Additionally, geomorphological and oceanographic datasets are readily available for these fjords.

Building on the work of Smeaton et al. (2016), which centred on Loch Sunart (56.705556, -5.737534), we focus on a further four fjords to develop site specific sedimentary C stock estimations, which then allow us to make more precise estimates for the same range of fjordic system types in Scotland. The chosen sites are Loch Etive (56.459224, -5.311151), Loch Creran (56.536970, -5.324578), Loch Broom (57.873974, -5.117443) and Little Loch Broom (57.872144, -5.316385)(Fig.1).These fjords differ significantly in their physical characteristics (Table 1) and bottom water oxygen conditions. Hypoxic bottom water conditions are recognised as an important factor in C burial and preservation within





depositional coastal environments (Middelburg and Levin, 2009; Woulds et al., 2007).
However of these, 111 fjords, only Loch Etive's upper basin is known to be permanently
hypoxic (Friedrich et al., 2014). Modelling of deep water renewal  in the 111 fjords suggests
that between 5 and 28 fjords, including Loch Broom and Little Loch Broom, could
experience intermittent periods of hypoxia, while this is less likely in Lochs Sunart and
Creran (Gillibrand et al., 2005, 2006).

**3. Towards A National Fjordic Sedimentary Carbon Inventory**

3.1. *Sample and Data Collection*

This study applies the methodology of Smeaton et al. (2016) where sediment cores and
seismic geophysical data were collected to four additional fjords. Figure. 1 shows the location
of each of the long (>1 m) sediment cores extracted from the four fjords chosen to produce
detailed sedimentary C stock estimates. With the exception of Loch Creran, where the
required data were extracted from  the available literature (Cronin  and Tyler. 1980, Loh et al.,
2008), each core was subsampled at 10 cm intervals for analysis. In total, 285 subsamples
were collected from the sediment cores from Loch Etive (n= 133), Loch Broom (n= 78) and
Little Loch Broom (n= 74). The data produced by Smeaton et al. (2016) for the glacially
derived sediment in Loch Sunart were used as a surrogate for all glacial sediments in this
study since MD04-2833 remains the only mid-latitude fjord core with chronologically
constrained glacial sediment (Baltzer et al. 2010). Detailed seabed seismic geophysical data
for Loch Etive (Howe et al., 2002) , Loch Creran (Mokeddem et al., 2015), Loch Broom
(Stoker and Bradwell, 2009) and Little Loch Broom (Stoker et al. 2010) was compiled.
In addition, sediment surface samples (n= 61) and partial seismic surveys (n=5) have been
collected from a number of additional fjords (Fig.1). These, in conjunction with data from the
literature (Russell et al., 2010, Webster et al., 2004), provide a greater understanding of C
abundance in these sediments and assist in constraining upscaling efforts. The full dataset is
presented in the supplementary material.

3.2. *Analytical Methods*

Each of the subsamples was split for physical and geochemical analyses. The dry bulk
density (DBD) of the sediment was calculated following Dadey et al. (1992). All samples
were freeze dried, milled and analysed for total carbon (TC) and nitrogen (N) using a Costech
elemental analysis (EA) (Verardo et al., 1990). Sub-samples of the same samples then
underwent carbonate removal through acid fumigation (Harris et al., 2001) and were analysed
by EA to quantify the organic carbon (OC) content. The inorganic carbon (IC) content of the
sediment was calculated by deducting the OC from the TC.  Analytical precision was
estimated from repeat analysis of standard reference material B2178 (Medium Organic
content standard from Elemental Microanalysis, UK) with C = 0.08 % and N = 0.02 % (n =
142   40).

3.3. *Fjord Specific Sedimentary Carbon Inventories*



Following the methodology of Smeaton et al. (2016), the geochemical and seismic
geophysical data were combined to make first order estimates of the C held in the postglacial
and glacial sediments of Loch Etive, Creran, Broom and Little Loch Broom. We then
calculated how effectively the fjord stores C ($C_{eff}$) as a depth-integrated average value per
$km^2$ for both the postglacial and glacial-derived sediments. Unlike Loch Sunart, where the
sediment stratigraphy has robust chronological constraints (Cage and Austin, 2010; Smeaton
et al., 2016), the four other fjords largely lack chronological evidence, with the exception of
two cores from Loch Etive (Howe et al., 2002; Nørgaard-Pedersen et al., 2006). The lack of
$^{14}$C dating means we rely solely on the interpretation of the seismic geophysics to
differentiate between the postglacial and glacial sediments. To ensure the consistency of this
approach, previous seismic interpretations of Scottish fjordic sediments (Baltzer et al., 2010;
Dix and Duck, 2000; Howe et al., 2002, Stoker and Bradwell, 2009, Stoker et al. 2010) were
studied and a catalogue of different seismic facies compiled for use as a reference guide
(Supplementary Material). Finally, we applied the framework set out in Smeaton et al. (2016)
to reduce uncertainty in the interpretation of the seismic geophysics by testing seismic units
against available dated sediment cores.
3.4. *Upscaling to a National Sedimentary Carbon Inventory*
Upscaling from individual to national coastal C estimates was key objective of this work.
Two approaches were developed to upscale the five detailed sedimentary C inventories to a
national scale stock assessment of C in the sediment of the 111 major Scottish fjords. Both
approaches utilise the physical characteristics of the fjords to quantify the OC and IC held
within the sediment. From these data we can also estimate the long-term average quantity of
C buried each year. Currently the best estimate of when the west coast of Scotland was free
of ice from the last glacial period is approximately 13.5 ka (Lambeck, 1993) though it could
be argued that 15 ka or 11.5 ka BP would be more appropriate. Modelling of the retreat of
the last ice sheet (Clark et al., 2012) suggests that a significant number of the fjords would
have been ice free around 15 ka (Supplementary Material) and have the ability to start
accumulating C. Alternatively 11.5 ka (Golledge, 2009) could be used as this date signifies
the point the fjords became permanently ice free after the loss of ice associated with the
Younger Dryas period. By dividing the total C held within the postglacial sediment in all the
fjords by this range of dates we can calculate the long-term average quantity of C buried per
year since the start of the postglacial period. Although the methodology is relatively crude
and probably underestimates the quantity of C being buried each year, it does give a valuable
first order insight into the long-term carbon sequestration service that fjords prove.
*3.4.1. Fjord Classification Approach*
The first stage of upscaling involves grouping the 111 fjords using the physical
characteristics identified in (Table. 1), along with rainfall, tidal range and runoff data.
Grouping was achieved by applying a k-means cluster analysis (1 x$10^5$ iterations) to all 111
fjords (Edwards and Sharples, 1986). This resulted in the delineation of four groups (Fig.2).
Group 1 comprises mainly mainland fjords which are the most deeply glaciated and have



highly restrictive submarine geomorphology (Gillibrand et al., 2005); Loch Sunart and
Creran fall into this category. Group 2 contains fjords from the mainland and the Inner
Hebrides which tend to be less deeply glaciated and more open systems; Loch Broom and
Little Loch Broom are part of this group. Group 3 includes the fjords on Shetland and the
Outer Hebrides; these fjords are shallower and their catchments tend to be smaller and
noticeably less glaciated. Group 4 consists of Loch Etive and Loch Linnhe; these fjords are
outliers from the other groups and both have extremely large catchments in comparison to the
others and were major glacial conduits for ice draining the central Scottish ice field at the last
glacial period. This analysis suggest the level to which the fjords are glaciated is a defining
factor to how they are classified. When mapped the ice thickness at the last glacial maximum
(Lambeck et al. 1993) largely correlates with the groupings produced by the k-means analysis
(Supplementary Material) with Group 1 under the maximum amount of ice, which reduces in
thickness for each subsequent group. Our case study fjords are thus representative of three of
the fjordic groups that can be recognised at a national scale. Group specific postglacial and
glacial $OC_{eff}$ and $IC_{eff}$ were calculated using the data from the detailed sedimentary C
inventories available from our five sites. The Group specific $OC_{eff}$ and $IC_{eff}$ were applied to
each fjord within a group, giving the total OC and IC stock for each fjord. Group 3 does not
contain any of the five fjords for which there are detailed C stock estimations and Group 2
has therefore been chosen as a surrogate since the k-mean analysis indicate that Groups 2 and
3 have the greatest similarities.
*3.4.2. Physical Attribute Approach*
The physical characteristics of fjords (Table 1) have primarily governed the input of C into
the fjord since the end of the last glaciation, when the majority of fjords became ice-free. We
might therefore expect a relationship between the physical features of a given fjord and its
accompanying catchment, and the C stored in its sediments. We use detailed sedimentary C
stock estimations in conjunction with the physical characteristics (Edwards and Sharples,
1986) to determine which physical feature best correlates with the quantity of OC and IC held
in the sediment. A statistical scoping exercise was therefore undertaken to determine which
physical characteristics are best suited to the upscaling process (Supplementary Material).
The results indicate that there are strong linear relationships between $OC_{eff}$ and tidal range ($p$
$= 0.012$, $R^2 = 0.909$), precipitation ($p = 0.003$, $R^2 = 0.961$), catchment area ($p = 0.023$, $R^2 =$
$0.860$) and runoff ($p = 0.019$, $R^2 = 0.877$). The correlation between these physical features
and OC content fits well with our understanding of fjord processes, since tidal range is a
proxy for the geomorphological restrictiveness of the fjord, while catchment size,
precipitation and runoff govern the input of terrestrially-derived OC (Cui et al., 2016) into the
fjord. The relationship between the IC stored in the sediment and a fjord's physical
characteristics is less well-defined, with strong correlations identified between IC and the
area of the fjord ($p = 0.009$, $R^2 = 0.925$) and the length of the fjord ($p = 0.016$, $R^2 = 0.892$).
Again, this fits with what we would expect: the larger/longer the fjord, the greater the
opportunity for in-situ IC production (Atamanchuk et al., 2015) and remineralisation of OC
(Bianchi et al., 2016) . Each of these relationships were used to calculate the OC and IC



stored in the postglacial sediment of each of the 111 fjords. The input of glacially-derived OC
during the retreat of the ice sheet at approximately 13.5 ka -17 ka(Clark et al., 2012) is
controlled by a more sporadic mechanisms (Brazier et al. 1988) governed by complex
advance-retreat ice margin dynamics during the deglaciation. This approach is therefore not
suitable for estimating the C stored in the glacial sediment of the fjords
*3.4.3. Constraining Estimates and Uncertainty*
To determine the accuracy of both upscaling methodologies, we compared the total quantity
of sedimentary OC and IC calculated for Lochs Sunart, Etive, Creran, Broom and Little Loch
Broom by both upscaling approaches alongside detailed estimates of C held within the
sediment of each of the five fjords. Although there are insufficient data to create additional
detailed sedimentary C stock estimates at a national scale, there are enough data from some
fjords to make broad estimations (Supplementary Data). Seismic geophysical data from
Lochs Hourn (57.125683, -5.589578), Eriboll (58.497543, -4.685106 ), Fyne (55.882882, -
5.381012), Nevis (57.007023, -5.693133) and Lower Loch Linnhe (56.591510, -5.456910)
allow us to estimate the minimum and maximum depth of postglacial sediment, while surface
sample data from each loch enables us to estimate C content of the sediment. Using these
data we can calculate basic estimates of postglacial OC and IC held within the sediment of
these fjords as an additional check on the accuracy of the upscaling methodology.
Two metrics of uncertainty were employed: arithmetic and a confidence-driven approach.
The arithmetic method follows the approach of Smeaton et al. (2016), whereby any known
arithmetic uncertainty is propagated through all the calculations. However, as recognised by
Smeaton et al. (2016), there are 'known unknowns' which we cannot reliably quantify.
Therefore we have further employed a confidence-driven approach to assess the final C stock
estimations for each fjord. Using a modified confidence matrix (Fig.3) following the
protocols adopted in the IPPC 5$^{th}$ Assessment (Mastrandrea et al., 2010), we have semi-
quantitatively assigned a level of confidence to the C estimates from each fjord. The matrix
uses the results from the k-means analysis and the availability of secondary data
(Supplemental Material) to assign a confidence level. For example, as described above (3.4.1)
a fjord in the Outer Hebrides would fall into Group 3. As discussed, this group is without a
detailed sedimentary carbon inventory and no other data are available t test the calculated C
inventory. In this case, the C stock estimation for that fjord would be assigned a very low
confidence level. In contrast, if the fjord fell into to Group 1, where there are similar fjords
with detailed C stock estimations and further C and partial geophysical data were available to
test the calculated C inventory, then a high confidence level is assigned. The five fjords with
detailed sedimentary C inventories are the only sites, which have been assigned a confidence
level of very high.
**4. Interpretation and Discussion**
*4.1. Fjord Specific Sedimentary Carbon Inventories*



Sedimentary analyses showed a broad similarity in dry bulk density values from the
postglacial sediment of the five fjords, while the variability between the fjords is more clearly
illustrated by the carbon data (Fig.4). Lochs Broom, Sunart and Little Loch Broom are
characterised by similar quantities of OC and IC. Although the TC content of the sediment in
Loch Creran is comparable to the other fjords, the relative contribution of OC is higher, with
a correspondingly lower quantity of IC in the sediment. Of the five lochs surveyed, the C
content of Loch Etive's sediment is significantly different from the other sites. It has the
highest TC content due to high quantities of OC found in the sediment. This is a possible
consequence of hypoxic conditions in the inner basin, as discussed below. As expected, the
highest dry bulk density values and lowest quantity of OC and IC occur in the glacial
sediments at all sites.
The total C held within each of the five fjords (Table 2) was calculated by combining the
bulk density data, % C and sediment volume models (Supplementary Material). Loch Sunart
($26.9 \pm 0.5$ Mt C) contains the largest sedimentary C store of the five fjords, closely followed
by Loch Etive ($21.1 \pm 0.3$ Mt C). In comparison, Lochs Creran, Broom and Little Loch
Broom hold significantly less C.  As indicated above, the postglacial sediments of Loch Etive
hold the greatest quantity of OC ($11.5 \pm 0.4$ Mt) with 7.76 Mt of that OC held in the upper
hypoxic basin resulting in Loch Etive being the most effective store of OC ($0.455$ Mt OC km$^{-2}$).
These results suggest that low oxygen conditions inhibit reworking and remineralisation of
organics and the production of carbonate fauna (Woulds et al. 2016) Loch Sunart has large
sills (Smeaton et al. 2016) and is one of the largest fjords in Scotland; these features favour
the storage of large quantities of post-glacial OC ($9.4 \pm 0.2$ Mt) and IC ($10.1 \pm 0.2$ Mt). The
quantities of C stored in the sediment of the smaller fjords are strongly linked to how
restrictive the geomorphology of the fjord is. For example, the smallest quantity of IC is held
within Loch Creran. This is in part be due to the shallow and narrow central sill which results
in a terrestrially dominated system with high sedimentation rates (Loh et al. 2008) which
increases the OC storage effectiveness ($0.195$ Mt OC km$^{-2}$) but reduces the IC storage
effectiveness ($0.068$ Mt IC km$^{-2}$) as increased humic acid input from terrestrial sources
(Bauer and Bianchi. 2011) results in lower pH which in turn reduces the suitability of the
fjord for calcifying organisms (Khanna et al. 2013). In contrast, the relatively unrestricted
geomorphology of Loch Broom results in the fjord being governed by marine processes
which creates a highly effective store of IC $0.232$ Mt IC km$^{-2}$ but in turn means these open
systems are comparatively poor at capturing OC as illustrated my Little loch Broom (1.6 Mt
OC).  The glacial material contains less C than the postglacial sediments. The effective
storage of C in the glacially-derived sediments of the five fjords is very similar, with the
OC$_{eff}$ ranging between 0.030 to 0.093 Mt OC km$^{-2}$ and an IC$_{eff}$ varying between 0.068 and
0.104 Mt IC km$^{-2}$ (Table 2). The similarity of these results may be because the mechanisms
governing the deposition of glacial sediment during the retreat of the British Ice Sheet
(Brazier et al. 1988) were similar across the geographic range of the fjords, but it may also be
a product of limited data availability for the glacial sediment.
*4.2 A National Fjordic Sedimentary C Inventory*



The results of the upscaling process suggest overall an estimated 640.7 ± 46 Mt C are stored
in fjordic sediments of Scotland, comprising 295.6 ± 52 Mt OC and 345.1 ± 39 Mt IC. The
postglacial sediments are the main repository for much of this C, with almost equal amounts
of OC and IC indicated by a OC:IC ratio of 1.17:1. In contrast, the glacial sediments are
dominated by IC, with an OC:IC ratio of 0.33:1. This is most likely due to the glacial source
material originating from scoured bedrock, and the absence of organic-rich soils and
vegetation (Edwards and Whittington. 2010.). The storage of C is unevenly distributed
between the 111 fjords; a small number of systems disproportionately contribute to the
national sedimentary C total (Fig.5). The sediment of fourteen large fjords hold 65 % of the
total C held Scotland's fjords (Table.4). Estimated C stocks for individual fjords can be found
in the supplementary material.

In addition to quantifying the total C stored in these fjords, we also calculated the accuracy of
the upscaling process (Supplementary Material) and assigned a confidence level to each of
the sedimentary C estimates) using the confidence matrix (Fig. 3). The availability of data for
the postglacial sediment means that we have medium to very high confidence in our estimates
of the quantity of OC and IC stored in 74 of the 111 fjords. The remaining 37 fjords have
been assigned a confidence level of low, with most originating from Group 3 of the k-means
analysis where we recognise a shortage of data needed to constrain C stock estimates.  The
lack of data for glacially-derived sediment results in all except the five case study lochs being
assigned a confidence level of very low to medium. Using these checks we believe that our
first order estimate of the C stored in the sediment of Scotland's fjords and the associated
uncertainties are realistic and robust. The confidence level assigned to each fjordic C estimate
stock can be found in the supplementary material.

### 4.2.1 National Estimates of C Burial

Annually an estimated 31139- 40615 t of C is buried in the sediment of the 111 fjords, with
OC contributing 16828 - 21949 t yr$^{-1}$ and IC supplying 14311 -18666 t yr$^{-1}$.  This annual
burial of C has been suggested to provide a climate regulating service through C
sequestration (Smith et al., 2015), yet efforts to fully quantify this mechanism have remained
elusive. The results from this study indicate that fjords have been capturing OC since the
retreat of the last ice sheet some of which that would have otherwise been lost to the open
ocean, where it would be more readily remineralized. Although the results do little to resolve
the mechanisms that govern this climate regulating service, they clearly show that fjords have
been providing this service since the retreat of the last ice sheet and throughout the Holocene.
This suggests that these systems have the capacity to adapt to changing environmental
conditions.  Intriguingly, there is also the possibility that this process may have aided the
capture of terrestrial C during the late Holocene and recent past (Smeaton and Austin, 2017,
submitted).

### 4.2.2 Global Outlook



Given similarities between the mid-latitude fjords and coastal environments of New Zealand,
Chile, Norway and Canada (Syvitski and Shaw, 1995), it is reasonable to suggest that our
findings are relevant throughout these systems. The sediments within fjordic environments
around the world potentially hold significant quantities of both OC and IC which have been
overlooked in national and global carbon budgets. The joint geophysical and geochemical
methodology used to quantify sedimentary C stocks coupled to the upscaling approach taken
in this study is capable of providing nations around the world with the ability to quantify of
their coastal sedimentary C stocks and reassess their nation's natural capital.
*4.3. Comparison to Other Mid-Latitude Carbon Stocks: significance and vulnerability*
The 640.7 ± 46 Mt of carbon held within the sediment of the fjords is one of the largest stores
in Scotland (Fig.6). The fjordic sedimentary store is the largest of Scotland's coastal carbon
stores (Burrows et al., 2014), exceeding both maerl and biogenic reefs which have been
shown to be highly effective stores of both OC and IC (Van Der Heijden and Kamenos,
2015). In addition, fjord sediments hold a greater quantity of C than all the living vegetation
in Scotland (Forestry Commission, 2015, Henrys et al., 2016, Vanguelova et al., 2013).
While Scotland's soils (Aitkenhead and Coull, 2016) and in particular the peatlands
(Chapman et al., 2009) contain a greater quantity of OC than the fjords, it must be
remembered that the fjord sediments also hold IC and the areal extent of these stores differs
greatly. When normalised by area (Fig.6), fjordic sediments emerge as a far more effective
store of OC and IC than other Scottish C stores, on land or at sea.
Globally, there are no direct comparisons as this is the first national C inventory of marine
sediments. Recent work in Denmark suggested that the Thurøbund seagrass meadow was one
of the most effective stores of C in the world, storing 0.027 Mt C km$^{-2}$ (Röhr et al., 2016). On
an aerial basis, however, these seagrass meadows are significantly less effective than fjord
sediments, which hold 0.219 Mt OC km$^{-2}$ and 0.256 Mt IC km$^{-2}$. This disparity emerges
because Röhr et al. (2016) only consider the top 0.25 m of seagrass sediment, while our study
encompassed the full depth of sediment. In Loch Sunart, for example, sediment depths of 70
m have been recorded (Baltzer et al. 2010). When compared like for like (i.e. the top 0.25 m)
the Thurøbund seagrass meadow is more effective at accumulating C, although questions
remain over the stability and longevity of these stores in comparison with the fjord sediments.
This is a key concern when comparing C stores.
Radiocarbon dating (Nørgaard-Pedersen et al. 2006, Baltzer et al. 2010, Smeaton et al. 2016)
shows that the fjords have been collecting sediment since the retreat of the last ice sheet
(Clark et al. 2012), which results in these C stores likely being some of the oldest and most
persistent in the UK. Of the terrestrial C stores, only soils and peatland have the potential to
store C over similar timescales, but they are significantly more vulnerable to natural and
anthropogenic disturbance than the fjordic sediments. Vegetation and soil C stores are at risk
from rapid and long-term environmental change. These environments can lose significant
quantities of C through soil erosion (Cummins et al. 2011), fire (Davies et al. 2013) and
vegetation change (Jackson et al. 2002), disturbances which are increasing in regularity and



severity with growing climatic and anthropogenic pressure. When we consider the marine sedimentary C stores through the same prism of environmental change, it is evident that the restricted geomorphology, water depth and relative remoteness of these stores affords them a level of protection not found in the terrestrial environment. However, this does not imply that coastal sedimentary C stores do not require careful management. For example, the remobilisation of C-rich sediments at the seafloor from direct physical disturbance poses an increased risk to these effective long-term C stores. The recognition of these coastal habitats for both their biodiversity and additional ecosystem functioning, including C sequestration and storage, represents an important emerging opportunity to designate and help create a new thinking in the establishment of marine protected areas. Taking into account the areal extent of fjords, their proximity to terrestrial sources and their longevity and stability, we suggest that fjordic sediments are the most effective systems for the long-term storage of OC in the UK and it is highly likely that fjords globally are just as effective as their mid-latitude equivalents at storing C.

## 5. Conclusion

The sediments of mid-latitude fjords hold a significant quantity of C which has largely been overlooked in global C budgets and which constitute a significant component of natural capital for Scotland and the UK. Our results indicate that the $640.7 \pm 46$ Mt C held within the sediments of these fjords is of similar, if not greater, magnitude than most terrestrial C stores. Fjords cover a small area in comparison with terrestrial C stores, but the stability and longevity of these coastal stores means that fjords are a highly effective long-term repository of C, surpassing the Scottish peatlands which have been the focus of intense research for decades. In contrast with their terrestrial equivalents, the magnitude of the fjord sedimentary C stores combined with their long-term stability emphasises the significant role that fjords and the coastal ocean, more generally, play in the burial and storage of C globally. This highlights the need for stronger international effort to quantify coastal sedimentary C stores and account for the C sequestration and associated climate regulating services which these subtidal environments provide.

**Author Contribution**

Craig Smeaton and William E. N. Austin conceived the research and wrote the manuscript, to which all co-authors contributed data or provided input. Craig Smeaton conducted the research as part of his PhD at the University of St. Andrews, supervised by William E. N. Austin, Althea L. Davies and John A. Howe.

**Acknowledgments**

This work was supported by the Natural Environment Research Council (grant number: NE/L501852/1) with additional support from the NERC Radiocarbon Facility (Allocation 1934.1015). We thank, the British Geological Survey (Edinburgh and Keyworth) for providing access to seismic profiles and access to sediment samples (Loan Number: 237389). The crew and captain of RV Calanus for assisting in sample collection finically supported by





the EU Framework V HOLSMEER project (EVK2-CT-2000-00060) and the EU FPVI
Millennium project (contract number 017008). Further seismic profiles and the CALYPSO
long core were acquired within the frame of the French ECLIPSE programme with additional
financial support from NERC, SAMS and the University of St Andrews. The authors would
like to thank Marion Dufresne's Captain J.-M. Lefevre, the Chief Operator Y. Balut (from
IPEV). Additionally, we would like to thank Colin Abernethy and Richard Abel (Scottish
Association of Marine Science) for laboratory support.



























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



**Figure Captions**

**Figure 1.** Map illustrating the location of Scotland's 111 fjords and the available data. Additionally, detailed maps present the sampling locations within (**D**) Loch Broom, (**E**) Little Loch Broom, (**F**) Loch Sunart (Smeaton et al., 2016), (**G**) Loch Creran (Loh et al., 2008) and (**H**) Loch Etive.

**Figure 2.** Output from the k-means analysis showing the spatial distribution of the four different groups of fjords.

**Figure 3.** Matrix depicting the relationship between data availability, similarity to modelled fjords and confidence level. Adapted from IPCC 5th Assessment Report (Mastrandrea et al. 2010).

**Figure 4.** Boxplots illustrating the (**A**) dry bulk density and (**B**) carbon content (%) compiled from the sediment cores extracted from the five fjords central to this research. Data for the glacially derived sediments collected from Loch Sunart (MD04-2833) are also presented.

**Figure 5.** Frequency distribution of sedimentary TC stock estimates for the Scotland's 111 fjords.

**Figure 6.** Comparison of the Scotland's national fjordic sedimentary C store other national inventories of C. (**A**) Carbon stocks (Mt) (**B**) Area of store (km$^2$) (**C**) Effective carbon storage (Mt C km$^{-2}$) for the 111 fjords. (**D**) Effective carbon storage (Mt C km$^{-2}$) for the other national C stores.



















**Table 1.** Key physical characteristics of each of the five fjords selected to produce detailed estimates of sedimentary C stocks.

| Fjord | Length (km) | Area (km²) | Mean Depth (m) | Max Depth (m) | Catchment Size (km²) | Fresh/Tidal Ratio |
|---|---|---|---|---|---|---|
| Loch Etive | 29.5 | 27.7 | 33.9 | 139 | 1350 | 120.4 |
| Loch Creran | 12.8 | 13.3 | 13.4 | 49 | 164 | 12.5 |
| Loch Broom | 14.7 | 16.8 | 27.3 | 87 | 353 | 14 |
| Little Loch Broom | 12.7 | 20.4 | 41.7 | 110 | 167 | 5.5 |
| Loch Sunart | 30.7 | 47.3 | 38.9 | 124 | 299 | 5.3 |




**Table 2.** Detailed sedimentary C stocks presented as total carbon (TC), organic carbon (OC)
and inorganic carbon (IC) held within postglacial (PG) and glacial (G) sediment of the fjords.
Additionally, we list the $C_{eff}$ for each fjord as a measure of how effectively the sediment
stores C.

| Fjord | | TC (Mt) | OC (Mt) | IC (Mt) | $C_{eff}$ (Mt C km$^{-2}$) | $OC_{eff}$ (Mt OC km$^{-2}$) | $IC_{eff}$ (Mt IC km$^{-2}$) |
|---|---|---|---|---|---|---|---|
| **Loch Etive** | | **21.1 ± 0.3** | **12.6 ± 0.3** | **8.6 ± 0.3** | **0.766** | **0.455** | **0.311** |
| | PG | 17.7 ± 0.4 | 11.5 ± 0.4 | 6.2 ± 0.3 | 0.639 | 0.415 | 0.224 |
| | G | 3.5 ± 0.2 | 1.1 ± 0.1 | 2.4 ± 0.2 | 0.127 | 0.040 | 0.087 |
| **Loch Creran** | | **4.8 ± 0.7** | **3 ± 0.5** | **1.8 ± 0.9** | **0.361** | **0.225** | **0.136** |
| | PG | 3.5 ± 0.6 | 2.6 ± 0.7 | 0.9 ± 0.4 | 0.268 | 0.195 | 0.068 |
| | G | 1.3 ± 0.9 | 0.4 ± 0.1 | 0.9 ± 1.2 | 0.098 | 0.030 | 0.068 |
| **Loch Broom** | | **6.8 ± 0.4** | **2.9 ± 0.4** | **3.9 ± 0.4** | **0.405** | **0.173** | **0.232** |
| | PG | 5.1 ± 0.5 | 2.4 ± 0.5 | 2.7 ± 0.4 | 0.304 | 0.143 | 0.161 |
| | G | 1.7 ± 0.3 | 0.5 ± 0.2 | 1.2 ± 0.3 | 0.101 | 0.030 | 0.071 |
| **Little Loch Broom** | | **7 ± 0.5** | **3.5 ± 0.5** | **3.5 ± 0.6** | **0.344** | **0.171** | **0.173** |
| | PG | 3 ± 0.7 | 1.6 ± 0.6 | 1.4 ± 0.8 | 0.148 | 0.078 | 0.070 |
| | G | 4 ± 0.3 | 1.9 ± 0.2 | 2.1 ± 0.4 | 0.196 | 0.093 | 0.103 |
| **Loch Sunart** | | **26.9 ± 0.5** | **11.5 ± 0.2** | **15.0 ± 0.4** | **0.560** | **0.243** | **0.317** |
| | PG | 19.9 ± 0.3 | 9.4 ± 0.2 | 10.1 ± 0.2 | 0.412 | 0.199 | 0.213 |
| | G | 7.0 ± 0.8 | 2.1 ± 0.3 | 4.9 ± 0.6 | 0.148 | 0.044 | 0.104 |





















**Table 3.** Total C stored in the sediment of Scotland's 111 fjords further broken down into the quantities of OC and IC stored is the postglacial and glacial sediments.

| | TC (Mt) | OC (Mt) | IC (Mt) |
|---|---|---|---|
| Postglacial | 467.1 ± 65 | 252.4 ± 62 | 214.7 ± 85 |
| Glacial | 173.6 ± 18 | 43.2 ± 12 | 130.6 ± 22 |
| **Total** | **640.7 ± 46** | **295.6 ± 52** | **345.1 ± 39** |





**Table.4.** Details of the fourteen fjords that disproportionately contribute to Scotland's Fjordic
Sedimentary C stock.

| Fjord | TC (Mt) | % of Scotland's Total of Fjordic Sedimentary C Stock |
|---|---|---|
| Loch Fyne | 99.70 | 15.56 |
| Loch Linnhe (Lower) | 92.28 | 14.40 |
| Loch Torridon | 30.82 | 4.81 |
| Loch Linnhe (upper) and Eil | 27.82 | 4.34 |
| Loch Sunart | 26.50 | 4.14 |
| Loch Ewe | 21.82 | 3.41 |
| Loch Etive | 21.11 | 3.29 |
| Long Clyde | 16.60 | 2.59 |
| Loch Hourn | 15.41 | 2.41 |
| Loch Ryan | 14.35 | 2.24 |
| Loch na Keal | 14.29 | 2.23 |
| Loch Nevis | 13.08 | 2.04 |
| Loch Scridian | 12.01 | 1.87 |
| Loch Carron | 10.52 | 1.64 |









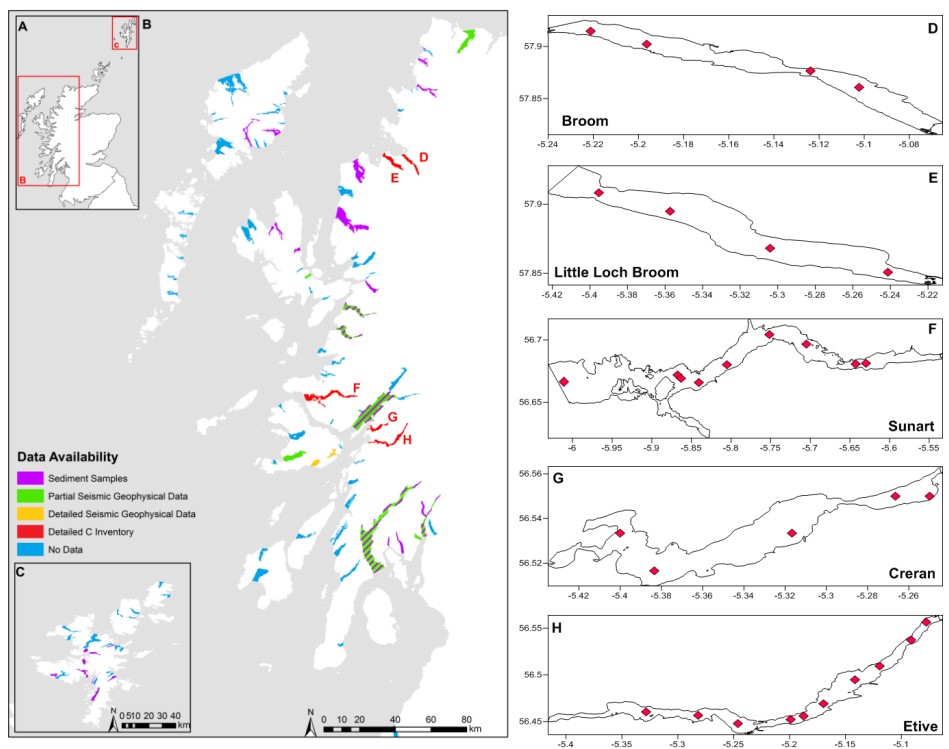




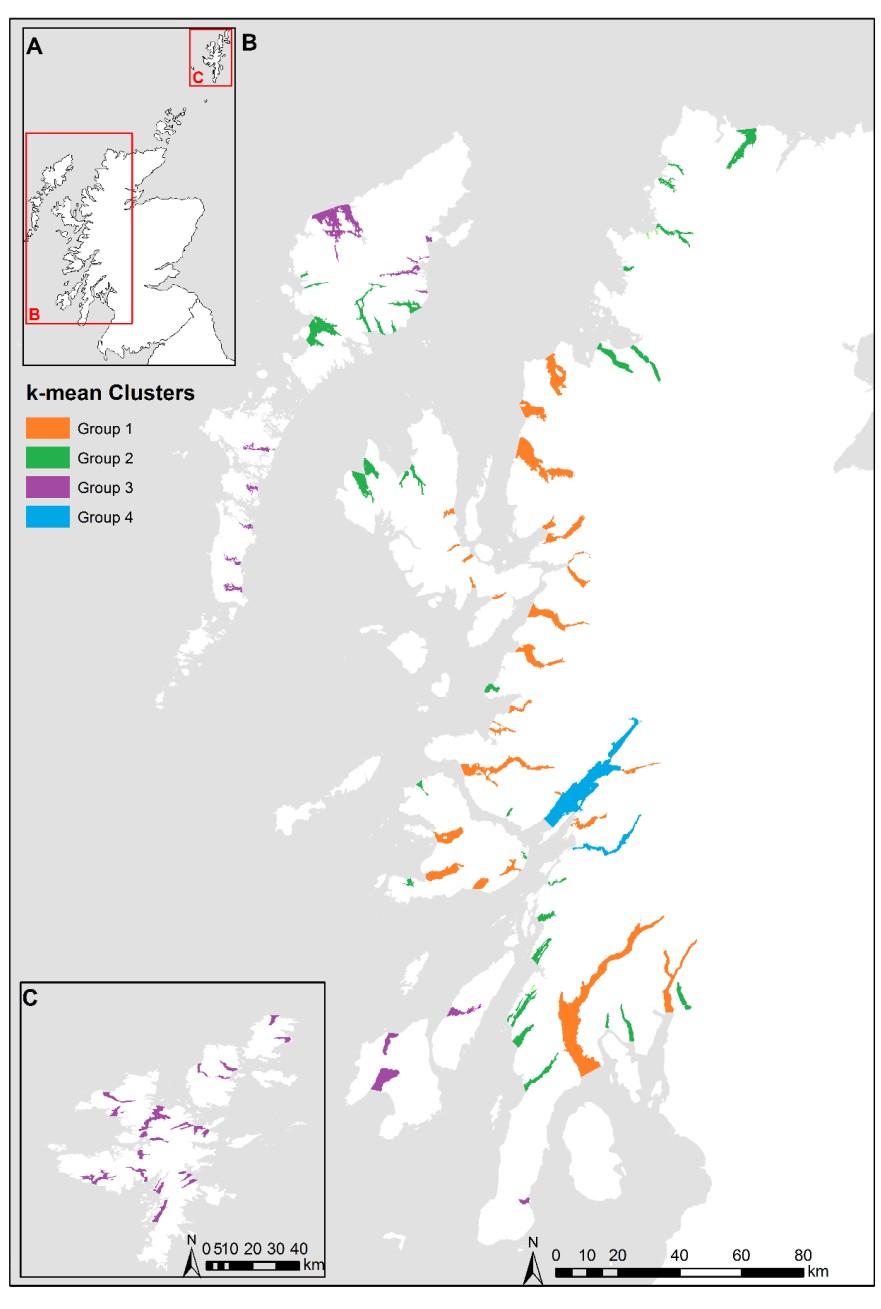












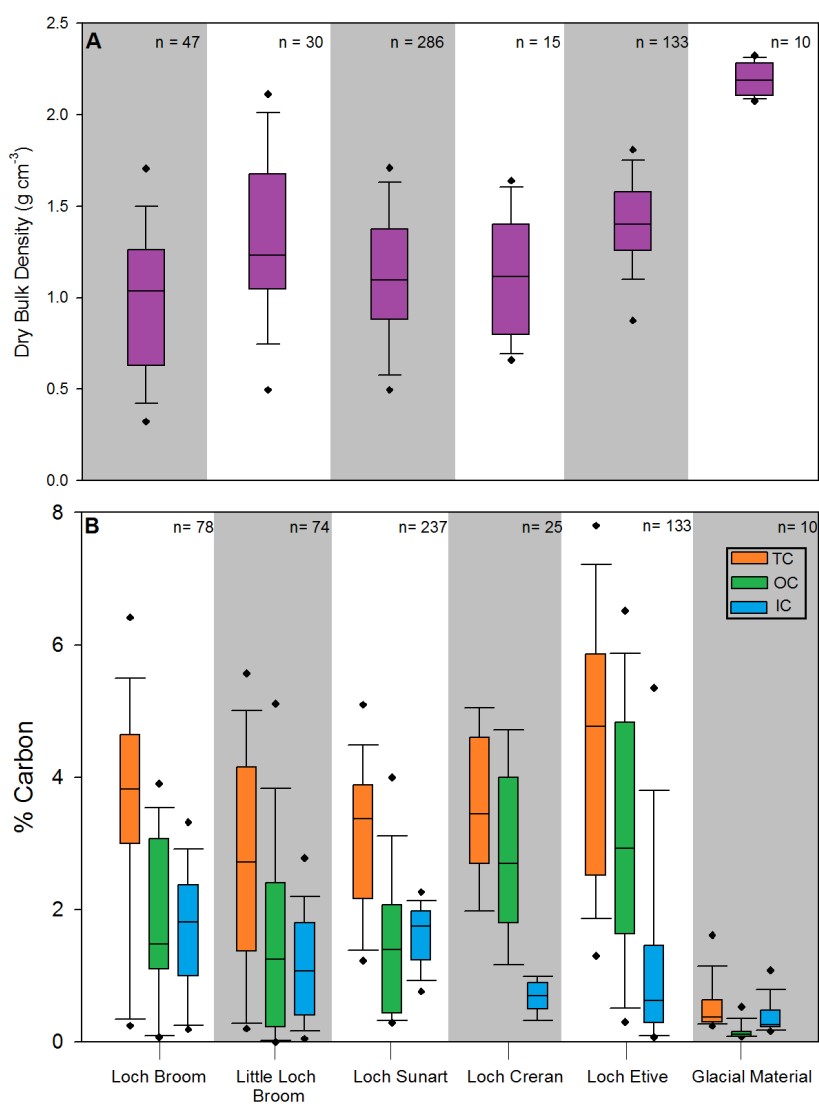















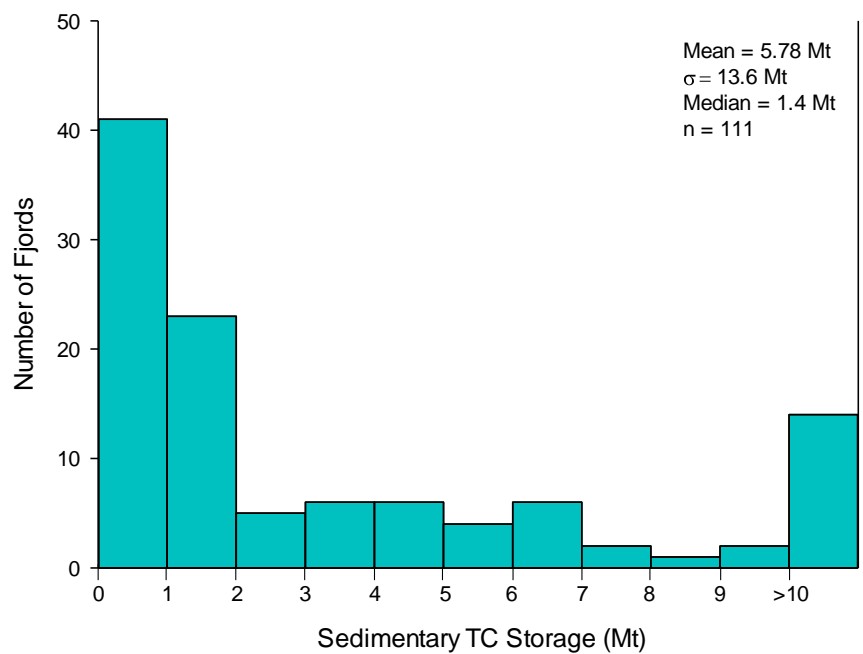












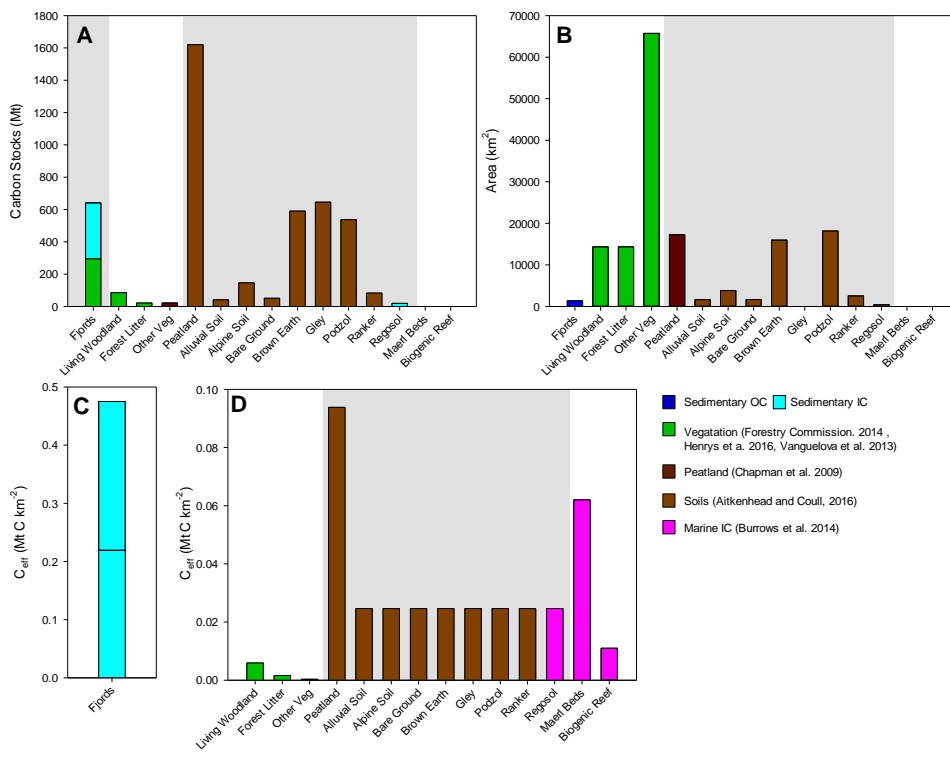








