# Peer review of "Scotland's Forgotten Carbon: A National Assessment of Mid-Latitude Fjord Sedimentary Carbon Stocks. Craig Smeaton 1,\*, William E.N. Austin 1,2, Althea L. Davies 1, Agnes Baltzer 3, John A. Howe 2 and John M. Baxter 4. 1 Sc"

_Biogeosciences, 2017_

## Referee Comment (RC1) · Anonymous Referee #1 · 27 Sep 2017

Smeaton et al. applied geochemical and geophysical methods to investigate the carbon stock in five representative fjords in Scotland and then used these five fjords using seismic and geochemical data and further modeled these five fjords. Results suggested strong similarity in estimated and calculated carbon stock numbers. They further applied this model to upscale to the national level and calculated the carbon budget in all Scotland fjords. This manuscript presented an interesting case study and also a valuable methodology advisable for future studies. I believe this manuscript is suitable for publication after minor revision. I only have one major concern about the manuscript, or maybe because I did not understand the methodology clearly, which requires further clarification. My understanding is that authors used seismic and car-

bon data to estimated carbon stock in these five fjords and then correlate them with parameters such as rainfall, catchment area, etc. These parameters were further used separately to calculate the carbon stock in each fjord. In my opinion, I believe it could generate a much more reliable number if the authors could incorporate all the parameters into one equation, such as carbon stock = a*precipitation*catchment area*runoff*tidal range. I am sure the equation could be further optimized based on the available data from these 5 fjords. This method has been largely used by Syvitski et al in modeling sediment discharge from global rivers. Besides that, I only have several minor comments: Line 180: change to .... Identified in Table 1. Line 206: a reference would be good. Line 224: as mentioned in the major comment and repeat again here: What if you combine all the parameters together, such as OC = a* tidal range*precipitate*catchment area*runoff. You could also modify the equation based on the best fitting. I think in this way, you could generate a more reliable OC and IC number. Line 254: . . . . . . . .. available to test. . . . . . . . Line 265: change carbon data to carbon concentrations? Lines 272-273: How do you conclude without glacial samples from all fjords? Line 283: If sills are a major reason affecting IC storage, then how it is possible to factor sills into the numeric model? Maybe I am confused here, but as it was mentioned earlier, the IC is modeled using fjord area and length. Line 295: change my to by Lines 334-336: any reference? Line 370: also depend on how deep is the seagrass habitat deposits Line 288: any reference?

---

## Referee Comment (RC2) · Anonymous Referee #2 · 6 Oct 2017

[General comments] This is the important study to estimate the national scale carbon stock of mid-latitude fjords. Although this estimation is a case study in Scotland, the methodology using seismic and biogeochemical data and the upscaling approaches are valuable and suggestive for estimating globally the carbon inventories of coastal waters. The upscaling methods contain uncertainties but the authors evaluate the uncertainties by IPCC protocol. I believe that this study is worth published in Biogeosciences. However, there are some points which should be addressed for publication. In my understanding, the authors compared the total quantity of sedimentary C calculated for five representative fjords by two upscaling approaches alongside detailed estimates of C stocks of each of the five fjords to check the accuracy of two upscaling methodologies. However, I cannot find any tables and figures about this point. I recommend adding a table or figure to certify the accuracy.

[Specific comments] Line180: What is Fresh/Tidal ratio in Table 1. How to calculate them? Line265: The large variation of %C in each fjord is found. Which is the dominant variation, vertical (sediment depth-related) or spatial variation? Line305: Please refer to "table 3". Line307: In postglacial sediments, the contribution of IC is similar to that of OC. What is the origin of IC? Line337: Please clarify the meaning of "changing environmental change". Line339: Please update the reference. If possible, please add the discussion about the origin of stored OC in the fjords. Figure 6: What is the meaning of shaded area? Is there any data in Maerl Beds and Biogenic Reef in Fig. 6A, B?

[Technical corrections] Line254: available "to" test Line295: by Little loch Broom? Line317: remove " ) " Figure 6: There are mistakes in the color of fjords. Vegatation -> Vegetation?

---

## Author Comment (AC1) · 16 Oct 2017

Smeaton et al. applied geochemical and geophysical methods to investigate the carbon stock in five representative fjords in Scotland and then used these five fjords using seismic and geochemical data and further modeled these five fjords. Results suggested strong similarity in estimated and calculated carbon stock numbers. They further applied this model to upscale to the national level and calculated the carbon budget in all Scotland fjords. This manuscript presented an interesting case study and also a valuable methodology advisable for future studies. I believe this manuscript is suitable for publication after minor revision.

[Figure]

*We thank the reviewer for the very helpful review, which highlights the significance of the national stock estimates and rigorous methodology adopted.

I only have one major concern about the manuscript, or maybe because I did not understand the methodology clearly, which requires further clarification. My understanding is that authors used seismic and carbon data to estimated carbon stock in these five fjords and then correlate them with parameters such as rainfall, catchment area, etc. These parameters were further used separately to calculate the carbon stock in each fjord.

*An Excel file detailing the statistical tests and results was has now been attached to the submission further detailing the methodology and providing greater clarity. We ask that this be included with the supplementary material; we make reference to this table in the revised manuscript text (lines 206-208).

In my opinion, I believe it could generate a much more reliable number if the authors could incorporate all the parameters into one equation, such as carbon stock = a*precipitation*catchment area*runoff*tidal range. I am sure the equation could be further optimized based on the available data from these 5 fjords. This method has been largely used by Syvitski et al in modeling sediment discharge from global rivers.

*The approach highlighted by the reviewer was undertaken. However, equations utilising all the parameters to determine C stock were highly variable and never produced C stock estimates comparable to the 5 fjords for which data was available. Several iterations of this equation were tested with little success (all of this is now included in the new supplementary table). We believe this numerical approach could be successful and could be used to further refine the these first order estimates but the lack of detailed C stock data is currently preventing its use; we have added a sentence (lines 320-322) to highlight this opportunity. We believe the methodology utilised in this manuscript is the best suited to produce a first-order national C stock estimate with the current data availability, but we recognise going forward refinement of these estimates
could use alternative numerical approaches as highlighted by the reviewer. This point, as noted above, is now acknowledged in the revised manuscript.

Minor comments: Line 180: change to Identified in Table 1. *Brackets have been removed.

Line 206: a reference would be good. *Reference added: McIntyre and Howe, (2010), Scottish west coast fjords since the last glaciation: a review, Geological Society, London, Special Publications, 344, 305-329, 1.

Line 224: as mentioned in the major comment and repeat again here: What if you combine all the parameters together, such as OC = a* tidal range*precipitate*catchment area*runoff. You could also modify the equation based on the best fitting. I think in this way, you could generate a more reliable OC and IC number. *See above comment.

Line 254: . . .. . .. . ... available to test. . .. . .. . .. *Typo corrected

Line 265: change carbon data to carbon concentrations? *Data changed to concentrations

Lines 272-273: How do you conclude without glacial samples from all fjords? *It is true that we only have glacial sediment samples from Loch Sunart and the data produced from these samples has been used to calculate the C stocks for glacial material in all 111 fjords. In Smeaton et al. (2016) we compared the C concentrations from the glacial marine sediments to glacial till deposited on land at the end of the last glacial period within the wider region. The C concentrations found in till compared well to that of the glacial marine sediment. Therefore we believe that C data from the Loch Sunart glacial samples is largely applicable to the wider network of fjords. We do accept there will be an error associated with these calculations which is reflected in the confidence level we have attributed to the calculations.

Line 283: If sills are a major reason affecting IC storage, then how it is possible to factor sills into the numeric model? *Though the sills are not directly used in the calculations,

the physical attributes of the fjords (Table 1) used in the calculations do reflect the role of the sills. The fresh/tidal ratio represents how restrictive the fjord geomorphology this is directly linked to the sill attributes.

Line 295: change my to by *Changed

Lines 334-336: any reference? *References Added: Bianchi, T. S.: The role of terrestrially derived organic carbon in the coastal ocean: a changing paradigm and the priming effect., Proc. Natl. Acad. Sci. U. S. A., 108(49), 19473–81, doi:10.1073/pnas.1017982108, 2011.

*References Added: Middelburg, J. J., Vlug, T., Jaco, F. and van der Nat, W. .: Organic matter mineralization in marine systems, Glob. Planet. Change, 8, 47–58, 1993.

Line 370: also depend on how deep is the seagrass habitat deposits *The depth of the seagrass sediments from Rohr et al. 2016 is unknown, this is the reason we do a like for like comparison (i.e. top 25 cm). The lack of fully depth integrated records is an issue for comparison is an issue highlighted in lines 363-373.

Please also note the supplement to this comment:
https://www.biogeosciences-discuss.net/bg-2017-360/bg-2017-360-AC1-supplement.zip

---

## Author Comment (AC2) · 16 Oct 2017

General comments This is the important study to estimate the national scale carbon stock of mid latitude fjords. Although this estimation is a case study in Scotland, the methodology using seismic and biogeochemical data and the upscaling approaches are valuable and suggestive for estimating globally the carbon inventories of coastal waters. The upscaling methods contain uncertainties but the authors evaluate the uncertainties by IPCC protocol. I believe that this study is worth published in Biogeosciences.

*We thank the reviewer for the very helpful review, which highlights the significance of

the national stock estimates and rigorous methodology adopted.

However, there are some points which should be addressed for publication. In my understanding, the authors compared the total quantity of sedimentary C calculated for five representative fjords by two upscaling approaches alongside detailed estimates of C stocks of each of the five fjords to check the accuracy of two upscale methodologies. However, I cannot find any tables and figures about this point. I recommend adding a table or figure to certify the accuracy.

*An Excel file detailing the statistical tests and results was has now been attached to the submission further detailing the methodology and providing greater clarity; as noted in response to reviewer 1 – a note in the revised text makes reference to this new, supplementary table.

Specific comments

Line180: What is Fresh/Tidal ratio in Table 1. How to calculate them? *The fresh/tidal ratio represents the ratio of supplies of fresh and tidal water as found in Edwards and Sharples (1986). Edwards and Sharples (1986) details the method used to calculate this ratio simply; fresh/tidal ratio = runoff/inflow

To add clarity Fig.1 caption now includes the reference to Edwards and Sharples (1986). Edwards, A. and Sharples, F.: Scottish Sea Lochs: A Catalogue. Scottish Marine Biological Association/Nature Conservancy Council, Oban, 1986.

*Line305: Please refer to "table 3". Reference to Table 3 has been added

Line307: In postglacial sediments, the contribution of IC is similar to that of OC. What is the origin of IC? *Generally the geology of the west coast of Scotland is igneous and metamorphic in nature therefore the main source of IC will be calcifying organisms (e.g. Foraminifera). In order to clarify the nature of the geological setting and its significance for sediment IC, a new sentence has been added into section 2 of the revised manuscript. Line 275-303 discusses that the fjords with highest IC content are also the

most marine influenced (fresh/tidal ratio), this paragraph highlights that the source of the IC are marine calcifying organisms.

Line337: "changing environmental change". *The sentence now reads: This suggests that these systems have the capacity to adapt to future environmental change

Line339: Please update the reference. If possible, please add the discussion about the origin of stored OC in the fjords. *Reference has been updated. Globally it has been estimated that approximately 66% of OC held in fjords is terrestrial in origin (Cui et al. 2016). In the Scottish context only one compressive study has taken place (Smeaton and Austin, 2017) which focused on Loch Sunart, it concluded that 44% of the OC held within the loch was terrestrial in origin. We believe that to fully include OC source contributions within this study would overstretch our current understanding of the Scottish fjord system and that this will require further and extensive field sampling (i.e. beyond the scope of this study). Line 285 has been updated to reflect our current understanding.

Figure 6: What is the meaning of shaded area? Is there any data in Maerl Beds and Biogenic Reef in Fig. 6A, B? *The shaded areas signify the broad environmental context. These shaded areas split the plot into three section 1) study results; 2) Living vegetation; 3) Soil; 4) Marine C stores. In order to simplify these plots, the background shading has been removed from the revised manuscript plot for Figure 6. Maerl and biogenic reef data are included in panels A and B of figure 6, but the total quantity of C stored in these environments and there areal coverage is small in comparison to the other data sets – this means that they do not readily appear on the plot (they are present). Only when normalized by area are they visible on the plot (i.e. panel C).

Technical corrections Line254: available "to" test *Typo has been corrected

Line295: by Little loch Broom? *My changed to by

Line317: remove " ) " *")" removed

Figure 6: There are mistakes in the color of fjords.

*The colouring of the fjords in figure 6 is consistent throughout. Panel B of figure 6 does not have the same two tone colouring as it is only referring to area not the OC and IC content. We believe the colouring of this figure is consistent and easy to follow. Vegatation -> Vegetation? Figure has been altered to vegetation

Please also note the supplement to this comment:
https://www.biogeosciences-discuss.net/bg-2017-360/bg-2017-360-AC2-supplement.zip